# HPV Infections—Classification, Pathogenesis, and Potential New Therapies

**DOI:** 10.3390/ijms25147616

**Published:** 2024-07-11

**Authors:** Beata Mlynarczyk-Bonikowska, Lidia Rudnicka

**Affiliations:** Department of Dermatology, Medical University of Warsaw, 02-091 Warsaw, Poland; lidia.rudnicka@wum.edu.pl

**Keywords:** HPV, systematics, pathogenesis, vaccines, pharmacological therapy

## Abstract

To date, more than 400 types of human papillomavirus (HPV) have been identified. Despite the creation of effective prophylactic vaccines against the most common genital HPVs, the viruses remain among the most prevalent pathogens found in humans. According to WHO data, they are the cause of 5% of all cancers. Even more frequent are persistent and recurrent benign lesions such as genital and common warts. HPVs are resistant to many disinfectants and relatively unsusceptible to external conditions. There is still no drug available to inhibit viral replication, and treatment is based on removing lesions or stimulating the host immune system. This paper presents the systematics of HPV and the differences in HPV structure between different genetic types, lineages, and sublineages, based on the literature and GenBank data. We also present the pathogenesis of diseases caused by HPV, with a special focus on the role played by E6, E7, and other viral proteins in the development of benign and cancerous lesions. We discuss further prospects for the treatment of HPV infections, including, among others, substances that block the entry of HPV into cells, inhibitors of viral early proteins, and some substances of plant origin that inhibit viral replication, as well as new possibilities for therapeutic vaccines.

## 1. Introduction

Human papillomaviruses (HPVs) are small, envelopeless viruses containing circular double-stranded DNA. More than 400 types of HPV are known, of which, depending on the database, more than 180 to more than 220 are fully classified. HPVs are among the most common pathogens affecting humans and genital HPV infection is considered the most common sexually transmitted disease. It is estimated that approximately one in ten sexually active women with normal cytology may be affected by current genital HPV infection. By the age of 45, the probability of HPV infection for sexually active individuals is assessed to be over 80% [1,2,3]. Although the majority of those infected remain asymptomatic and eliminate the infection, some individuals may experience persistent or recurrent benign lesions, while others may develop precancerous lesions and cancer. According to the WHO, HPV infections are responsible for approximately 5% of all cancers worldwide, and every year 625,600 women and 69,400 men develop cancer due to HPV infection [4]. Almost all cases (99.7%) of cervical cancer are caused by HPV and 80% of all cancers caused by HPV are cervical. However, HPV infections are also responsible for a large proportion of anal (71–90%), vaginal (65–74%), penile (43–63%), vulvar (43–74%), and head and neck cancers (10–70%) [5,6].

## 2. A Brief Historical Overview of HPV Research

The first descriptions of genital and cutaneous warts appeared as early as ancient times (i.e., by Hippocrates and by Celsus). In 1842, Italian physician Rigoni-Stern published a study in which he showed that cervical cancer is more common in sexually active women. This gave grounds to look for a link between cervical cancer and sexually transmitted diseases as early as the 19th century. However, no such correlation was found until the second half of the 20th century. In 1891, Payne demonstrated the infectious nature of common warts and in 1907 Ciuffo showed the viral etiology of these lesions. In 1949, Strauss and others found HPV particles in skin warts using electron microscopy. A series of studies conducted by Rous and subsequently by Syverton in the 1930s and early 1950s, using the cotton-tailed rabbit as a model, demonstrated an association between CRPV infection and the development of cancer in these animals. In 1957, the viral etiology of lesions occurring in epidermodysplasia verruciformis (EV) was demonstrated and in later years the link between HPV infection and the formation of skin cancers in the course of this disease was confirmed. The research was conducted, among others, by Jablonska, who headed our clinic at the time. In 1965, Crawford, Klug, and Finch described the structure of HPV ds DNA isolated from skin warts. In the early 1970s, Zur Hausen put forward the hypothesis that cervical cancer is caused by HPV infection; he later received a Nobel Prize in 2008 for proving this [7,8].

## 3. Classification of HPV

The name human papillomavirus covers the papillomaviruses (PVs) found in humans. The classification of PVs is based on the analysis of differences and similarities in the viral DNA sequence. Of particular importance is the sequence encoding the L1 capsid protein, for which the similarity between subfamilies should be no more than 45% and between genera no more than 60%. However, in practice, a phylogenetic algorithm based on the comparison of L1, L2, E1, and E2 sequences, and sometimes sequencing of the whole viral genome, is often used. New types of PVs should differ in their L1 gene sequence from previously known types by at least 10%. If the L1 sequence differs by 2–10%, we can speak of subtypes, and if it differs by less than 2%, we speak of variants. For grouping genetically similar viruses within a type, the terms lineages and sublineages are also used (e.g., in GenBank). Accordingly, animal papillomaviruses are classified into the same genera as human papillomaviruses, and the genetic similarity between some human and animal papillomavirus types may be greater than between some human-infecting types. The most clinically important HPVs, including high-risk mucosal HPVs such as HPV 16, 18, 26, 31, 33, 35, 39, 45, 51, 52, 53, 56, 58, 59, 66, 68, 73, 82 and low-risk mucosal HPVs such as HPV6, 11, 40, 42, 44, 54, 55, 61, 62, 71, 74, 81, 84, 89 (CP6108), 90 as well as skin-wart-causing HPVs such as HPV 1, 2, 3, 7, 10, 27, 57, 73, are classified as alphapapillomaviruses. Betapapillomaviruses include HPVs associated with epidermodysplasia verruciformis (EV) like HPV 5 and 8 [9,10,11]. Table 1 shows the systematics of HPVs according to the International Committee on Taxonomy of Viruses [12], which were compared with the contents of various databases [13,14,15]. The lineages and sublineages of HPV types based on the literature and GenBank accession no. of each HPV are also included. The table includes HPV types categorised into 49 species within 5 genera. An expanded version of Table 1 including links to GenBank can be found in Appendix A.

There are numerous distinctions between the ICTV and the Taxonomy Browser, IRHC, and PaVE databases. First of all, they differ in the number of HPV types classified. In the ICTV [12] database, there are 183 HPV types (numbering 1–45; 47–54; 56–63; 65–78; 80–175; 178–180; 184; 187; 197; 199–202; 204–205), while 204 HPVs (numbering 1–45, 47–63, 65–78, 80–182, 184–189, 191–193, 195–197, 199–202, 204–205, 209–210, 230) and HPVRTRX7, HPVXS2, HPVV001/Slovenia/2010, and HPVFA75/KI88-03 are classified in the Taxonomy Browser [13]. Unclassified Gammapapillomaviruses in this database include HPV190, 194, and KC5. Unclassified Papillomaviridae include HPV64, 198, 203, 211–216, 219–222, 226, 228, 229, MM8, HANOA 464, JC9710, JC9813, JEB2, and me180, Xc, Xd, Xf, Xg, Xh, AZ1_1, mSD2, and mSK_220.

There are 225 HPV types in the IRHC [14] database (numbering 1–45, 47–54, 56–63, 65–78, 80–216, 219–231) and 199 classified HPVs in the PaVE [15] database (numbering 1–45, 47–54, 56–63, 65–78, 80–156, 159–202, 204, 228, 229). HPV157, 158, 203, 205, 207–208, 210–216, 219–225 occur as unclassified Gammapapillomaviruses. Unclassified Betapapapillomaviruses include HPV206 and 209.

In addition, there are numerous differences in the classification of the different types. In the IRHC [14] database, HPV46, 55, 64, 79, 217, 218 do not appear, because they have been reclassified to HPV20, 44, 34, 91, 182, 189, respectively. In ICTV [12] and PaVE [15], they do not appear. According to the TB database, HPV55 belongs to the species Alphapapillomavirus 10 and HPV64 belongs to unclassified Papillomaviridae. HPV158, according to TB [13] and ICTV [12], is classified as Gammapapillomavirus 12 and, according to IRHC, is classified as Gammapapillomavirus 1, and in the PaVE [15] database occurs as unclassified Gammapapillomavirus. According to the IRHC database, HPV230 is classified as Gammapapillomavirus 15 and according to TB [13] it is classified as Gammapapillomavirus 11 and in PaVE [15] and ICTV [12] it does not occur. Only the TB database [13] includes HPVXS2 (alphapapapillomavirus 2), HPVV001/Slovenia/2010 (betapapapillomavirus 1), and HPVFA75/KI88-03 (betapapapillomavirus 2). In ICTV, IRHC, and PaVE, these types are not present.

Viruses classified in IRHC [14] and in PaVE [15] as HPV1 and HPV206 occur in TB as HPV1a and HPVRTRX7, respectively. HPV1 is present in ICTV [12] and HPV206 is absent. HPV198 (Betapapillomavirus 2), HPV214 and HPV226 (Gammapapillomavirus 6), HPV203 and HPV229 (Gammapapillomavirus 7), HPV211 (Gammapapillomavirus 8), and HPV215 and HPV216 (Gammapapillomavirus 9) are present in IRHC [14]; HPV221 (Gammapapillomavirus 10), HPV212 and HPV220 (Gammapapillomavirus 17), and HPV222 (Gammapapillomavirus 19) do not occur in ICTV [12] and are classified as unclassified HPV in the TB [13] database. Occurring in IRHC [14] and PaVE [15], HPV194 (Gammapapillomavirus 20) and HPV190 (Gammapapillomavirus 24) are not present in ICTV [12] and are present as unclassified Gammapapilomavirus in the TB [13] database.

HPV227 (Gammapapillomavirus 2), HPV225 (Gammapapillomavirus 7), HPV224 (Gammapapillomavirus 8), HPV231 (Gammapapillomavirus 10), and HPV223 (Gammapapapillomavirus 22) are not present in the IRHC [14] database in ICTV [12] and are present as unclassified Gammapapapilomavirus in the PaVE [15] database, with the exception of HPV231, which is not present in the database. In the TB [13] database, these viruses appear as unclassified HPV isolates ICB2, MTS4, ICB1, CDCHPVTL_S18, and MTS3.

## 4. General Characteristics and Structure of HPV

Human papillomaviruses (HPVs) are small (55 nm in diameter) non-enveloped, epitheliotropic viruses that carry double-stranded, circular DNA containing from 7100 bp (HPV48; GenBank acc.no. NC_001690) to 8104 bp (HPV83, GenBank acc.no. AF151983). HPV18 variants contain 7824 bp to 7857 bp (GenBank acc.no. KC470225, AY262282) and HPV16 variants contain 7885 bp to 7909 bp (GenBank acc.no. HQ644257, HQ644298).

HPVs have an icosahedral capsid consisting of 72 capsomers. A single HPV virus capsomer consists of five L1 proteins of 55 kDa. Variable copies but less than 72 molecules (typically 20–40 molecules) of minor capsid protein L2 are incorporated within the viral particle. The estimated molecular mass of HPV L2 is approximately 55 KDa. However, due to some posttranslational modification, L2 typically exhibits an apparent molecular weight of 64–78 kDa as determined by SDS-PAGE analysis [43,116,148,149,150,151,152,153].

The structure of the capsid and the lack of an envelope give HPVs low susceptibility to external agents. HPV is transmitted through direct contact and through certain objects. Mucosal HPV is mainly transmitted by sexual contact but there is also the possibility of extra-sexual infection. Vertical transmission is possible (from mother to child in the womb and more often perinatally). Studies in an animal model (mouse tail) have shown that HPV can remain on the fomite surface and retain infectivity for up to 8 weeks, and up to 1 year in exfoliated epithelial cells. During transfer from fomite to another surface, the viral titer drops by 10 times, which significantly reduces infectivity. It is worth noting that disinfectants containing ethyl alcohol, isopropyl alcohol, or octenidine are ineffective. HPVs are also resistant to desiccation, retaining 30% infectivity after 7 days of dehydration. However, these viruses are susceptible to ortho-phthalaldehyde and hydrogen peroxide, as well as to UVC [154,155,156].

### HPV Genome

The HPV genome can be divided into three regions: E (early), L (late), and LCR (long control region). The E region encodes early proteins (E1–E2, E4–E7 genes) responsible for replication (E1 and E2 genes), transcription (E2 gene), viral release (E4 gene), and creation of an environment favorable to virus replication including evasion of the host immune system and maintenance of epithelial cell proliferation (E5, E6 and E7 genes). As a result of transcription, additional proteins E6^E7, E1^E4, and E8^E2 (E8 is a fragment of E1) are formed from intragenic promoters [157]. The E8^E2 protein can functionally replace the E5 protein [106]. The L region encodes the L1 and L2 capsid proteins, while the non-coding LCR (long control region) contains cis regulatory elements and is involved in the regulation of viral replication and viral gene expression. Depending on the HPV type, the viral genome contains 5–11 ORFs (open reading frames). The HPV108 genome encodes only 5 proteins: E7 (99 aa), E1 (626 aa), E2 (390 aa), L2 (517 aa), and L1 (513 aa), while in the HPV41 genome 11 proteins are encoded: E6 (156 aa), E7 (114 aa), E1 (614 aa), X (77 aa), E2 (387 aa), Y (72 aa), E4 (101 aa), E5 (78 aa), L2 (554 aa), L1 (583 aa), and Z (76 aa). Atypical ORFs encoding X (100 aa) and Y (107 aa) proteins are also found in HPV81, those encoding L3 protein (110 aa) in HPV5b, those encoding E10 protein (37 aa) in HPV101, and those encoding two L1 proteins (504 aa and 532 aa) in HPV3. In the Alphapapillomavirus genus, 7–9 ORFs are most common (in HPV16 and HPV18, there are eight ORFs: E7, E6, E1, E2, E4, E5, L2, L1). The structure of the HPV genome, using HPV16 as an example, is shown in Figure 1.

It is worth noting that the genes encoding E1, E2, and L2 are the most conserved. L1 is more variable than L2. The remaining OTFs are characterized by much higher variability. This variability in the case of E5–E7 may, among other things, determine the oncogenic potential of the viruses and, in the case of E4, also the affinity for specific regions or the route of transmission [159]. Protein variants occurring in different types and lineages within a type can affect not only pathogenicity including oncogenicity and affinity to specific localizations but also immune response and the effectiveness of prophylactic and therapeutic vaccines and potential drugs. A summary of the major HPV types’/subtypes’ genome size and the number of amino acids in the E1–E2, E4–E7, and L1–L2 proteins is shown in Appendix A.

## 5. HPV Replication Cycle

The gateway to infection is a wound or micro-damage to the epithelium, which allows the virus to reach the basal layer. HPV can only infect dividing keratinocytes of the basal layer, e.g., in the healing process of an injury. Initially, the major capsid protein L1 binds to laminin-332 in the basement membrane. Next, the L1 protein is cut off by kallikrein-8 (KLK8), leading to a change in its conformation. The next step is the fusion of L1 with heparan sulfate proteoglycans (HSPGs) on the cell surface of the epithelial or epidermal basal layer. Further conformational change of L1 and L2 associated with the interaction between positively charged lysine residues in L1 and negatively charged HSPGs and then with the action of cyclophilin B (CyPB) leads to the exposure of the N-terminus of the L2 capsid protein. This allows the exposed L2 fragment to be cleaved by furin, leading to a decrease in the affinity of the capsid protein for HSPGs. Subsequently, L2 binds to the S100A10 subunit of the annexin A2 heterotetramer on the cell membrane, leading to clathrin-independent endocytosis of HPV into the cell [160,161]. The results of some studies suggest that HPVs may differ in the ways they enter cells. The penetration of HPV 18, 31, and 45 but not HPV 16 is dependent on glycosaminoglycans (GAGs) and can be inhibited by carrageenan, a polysaccharide derived from algae containing 15–40% ester-sulfate that blocks the interaction between viruses and GAGs [162].

At further stages, viral DNA is released from the capsid with the participation of cyclophilins, although the disassembly of the capsid is probably not complete. The L1–L2–viral DNA complex binds to cytoplasmic trafficking factors and is transported through the structures of the Golgi apparatus toward the cell nucleus, into which it enters using the moment of nuclear membrane disintegration at the beginning of mitosis. Vesicles containing viral DNA, L2, and probably L1 are then transported along microtubules, connect to mitotic chromatin, and at the end of mitosis become part of newly formed promyelocytic leukemia nuclear bodies (PML NBs) which protect the viral DNA from destruction by host restriction enzymes. The initiation of viral gene transcription is likely to be influenced by interactions with some PML NB-forming proteins, including Sp100. The first to be expressed are the HPV *E1* and *E2* genes which, among other things, leads to the activation of viral DNA replication [162].

The E1 protein has helicase activity and forms a complex with E2, resulting in increased specificity to the target DNA sequence. In infected cells of the basal layer, only a small number (about 50–100) of copies of the HPV genome are produced (initial amplification), then these copies are duplicated during cell division (maintenance amplification). One infected cell remains in the basal layer, while other cells continue to enter the suprabasal layer. A large number (thousands per cell) of virus copies are produced, only as the epithelial cells differentiate, in the upper layers of the epithelium (vegetative amplification) [161,163,164].

E2 is in the first place involved in the regulation of viral gene expression, including E5, E6, and E7. Expression in and above the basal layer of E5, E6, and E7 creates the right conditions for HPV replication by, among other things, stimulating cell proliferation, inhibiting cell apoptosis, and evading the host immune response. Unfortunately, similar conditions can also promote the development of precancerous conditions and cancer. A more detailed description of these proteins can be found in the section on carcinogenesis.

The L1 and L2 genes are expressed in terminally differentiated cells in the upper layers of the epithelium, which are associated with encapsidation. Late proteins for HPV16 are synthesized from the p670 promoter and for HPV18 from the p811 promoter [43]. The expression of late HPV genes is regulated by a variety of mechanisms including polyadenylation, alternative splicing, and may also be regulated at the translational level. The subsequent release of viral particles is mediated by the E4 protein, which disrupts the cellular network of cytokeratin filaments. E4 also causes cell cycle arrest and is involved in the assembly of virions [165,166,167]. The life cycle of HPV is shown in Figure 2.

## 6. Mechanisms of Precancerous and Neoplastic Lesions in HPV Infection

Early viral proteins such as E5, E6, and E7 play the most important role in the pathogenesis of neoplastic lesions arising from HPV infections.

The E5 protein is responsible for the control of cell growth, differentiation, and immune modulation. It is expressed in the early phase of replication. Small, hydrophobic, single E5 proteins are formed, which diffuse across the membrane, bind to platelet-derived growth factor receptors (PDGFR) and epidermal growth factor receptors (EGFR), enhancing their signaling, leading to the inhibition of apoptosis and stimulation of proliferation. The E5 protein is present in all high-risk mucosal HPV types; it promotes tumor progression as an early oncoprotein, but it does not occur in Betapapillomaviruses, including those classified as possibly carcinogenic (HPV5 and 8)—EV-associated HPVs [168,169].

The E6 protein (150–160 amino acids, about 18 kDal) is a major oncoprotein. It inhibits apoptosis and differentiation. It affects cell shape, polarity, mobility, and signaling [170]. The E6 protein binds to the cellular anti-oncogene p53, leading to its inactivation. In the case of oncogenic HPV, like HPV16, E6 binds to the ubiquitin ligase E6AP at a site of the latter’s short amino acid sequence LxxLL. The E6–E6AP complex then binds to p53, leading to the degradation of p53 by the ubiquitin-related system. E6 binds to p53 at a specific site, different from DNA or cellular protein attachment sites, which means that E6 can bind to both free and bound p53 [171,172]. One study found that the ability to degrade p53 through the ubiquitin-binding system was also possessed by HPV71, which is classified as non-oncogenic. This indicates that this feature alone does not determine the oncogenicity of the virus and that several mechanisms need to act simultaneously [173].

E7 binds to the retinoblastoma tumor suppressor protein pRb and to smaller proteins such as p107 and p130, inhibiting their action and, in the case of oncogenic HPV, leading to accelerated degradation. This results in the activation of the pRb-blocked transcription factor E2F, which in turn causes the infected cell to enter S phase (DNA replication). E7, only in the case of oncogenic HPVs, furthermore binds to the non-receptor protein tyrosine phosphatase PTPN14, leading to its degradation, which in a pRb-independent mechanism inhibits keratinocyte differentiation and promotes their immortalization [174,175].

E6 and E7 may also interact with other metabolic pathways involved in cell differentiation and proliferation. An important role in cell immortalization is played by the activation of telomerase by E6 and, to a lesser extent, E7 proteins produced by oncogenic HPV. The viral proteins act by increasing the expression of the catalytic subunit of telomerase, so-called human telomerase reverse transcriptase (hTERT) [165,176]. Oncogenic HPVs’ E6, unlike non-oncogenic ones, have a PDZ-binding motif (PBM) on their carboxy terminus. PDZs are a group of 80–90 amino acid domains named after the first three proteins discovered to possess them: Post Synaptic Density 95 (PSD95), the Discs Large (Dlg), and the Zona Occludens 1 (ZO-1). This group contains, among other things, proteins like DLG1, SCRIB, and MAGI1/2/3 that influence cell polarity—the asymmetric spatial organization of cell structures found, for example, in epithelial cells. Other proteins containing this domain can also influence cell shape (including by affecting tight junctions) and numerous signal transduction pathways including TGF (transforming growth factor)-β and PI3K (phosphatidylinositol-3-kinase)/AKT signaling. The association of E6 with PDZ may play roles in sustaining epithelial cell proliferation and the HPV replication cycle and also in tumor transformation. In addition, E6 has the ability to stimulate the Wnt/β-catenin and Notch metabolic pathways. Among other things, E6 and E7 proteins may also activate the intracellular signal transduction pathways PI3K/AKT/mTOR and JAK/STAT. The metabolic pathways listed above play a role in the pathogenesis of many cancers [170,177,178].

In most cases of HPV infection, the DNA remains in an episomal form, unintegrated into the host DNA, allowing the virus to replicate efficiently. It has been postulated that the integration of viral DNA into cellular DNA may be an indirect consequence of the entrapment of HPV episomes within cellular chromosomal structures. The integration appears in the course of persistent HPV infection, and causes the inhibition of the HPV replication cycle. Persistent infection affects no more than 10% of individuals infected with oncogenic HPV. The integration of the HPV genome into the cellular genome has been demonstrated to play a crucial role in the development of cancer. Studies have shown that the site of integration, as well as the size and region of the viral DNA, can vary among individuals. Changes in the number of copies of the genetic material and rearrangements within and between chromosomes are also possible. Loss of ORF for E2, a protein acting as a repressor for E6 and E7, has been described. As a result, the oncoproteins E6 and E7 are overexpressed [178,179].

It is worth noting, however, that integrated HPV 16 DNA was detected in about 74% and not in all cases of cervical cancer caused by HPV 16, and in head and neck cancers the percentage was even lower, suggesting that integration of viral DNA is not a prerequisite for carcinogenesis associated with HPV infection. Recently, a role for an alternative pathway of carcinogenesis has been suggested, in which HPV16 remains in an episomal form and there is increased expression of E2/E4/E5. The activation of EGFR by E5 can lead to activation of the receptor tyrosine kinase c-met, with potential oncogenic effects. E5 also inhibits EGFR degradation, increasing the presence of this receptor on the cell surface. In oropharyngeal cancer, the activation of fibroblast growth factor receptor (FGFR) and possibly also mTOR kinase has also been demonstrated. A better understanding of the above mechanisms may allow more effective treatment of HPV infections and the cancers they cause [180].

## 7. Immune Response and Immune Evasion by the Virus in HPV Infections

HPV infections are usually limited to the epidermis or mucosal epithelium; viral particles do not enter the blood and the replication cycle does not involve cytolysis, which reduces contact with the host immune system and therefore the immune response to infection. On the other hand, the epidermis and mucosal epithelium form an effective mechanical barrier and only when they are damaged can HPV infection occur [181].

It has been shown that the production of specific antibodies is of primary importance in the prevention of HPV infection, whereas cell-mediated immunity plays a greater role in the eradication of an existing infection or the resolution of lesions. In the course of a natural infection, antibodies are produced in 50–80% of women and 2–51% of men, usually after 6–12 months of infection. The concentration of antibodies is usually low, and they do not play a major role in the immune response against HPV infection. In contrast, prophylactic HPV vaccines induce a much higher concentration of antibodies and thus effectively prevent infection [182].

Persistent and recurrent HPV infection is favored in those with congenital or acquired cellular immunodeficiency, e.g., in HIV-infected patients and those taking immunosuppressive drugs such as transplant recipients. Such individuals are more likely to be infected with both the mucosal and cutaneous types of HPV and, in the case of oncogenic HPV infections, precancerous and neoplastic lesions are more likely to develop [183].

Innate and adaptive immunity are important in HPV clearance. HPV genetic material is recognized by toll-like receptors TLR 2, 3, 7, 8 and 9, resulting in increased production of interferons α and β. HPV DNA also acts on other pattern recognition receptors (PRRs) such as absent in melanoma 2 (AIM2), which leads to increased production of caspase and interleukin 1, and on interferon-γ (IFN-γ)-inducible protein 16 (IFI16), which restricts replication and transcription of HPV genes [181]. Also important in the early response to HPV infection are natural killer (NK) and natural killer T (NKT) cells. However, the most important role in the resolution of existing lesions appears to be played by cellular adaptive immunity. Specific CD 8+ and CD4+ T-cell infiltrate is found in spontaneously resolving HPV lesions. The specific response is directed against various HPV antigens, but especially against the E6, E7, and E2 proteins. Approximately 90% of all HPV infections are eliminated by the immune system and this process has been studied quite extensively. For E6 and E7 of HPV 16 and 18, the peptide sequences of the individual epitopes are known to stimulate an effective cytotoxic T-lymphocyte response against HPV infection. These sequences can be used in the development of therapeutic vaccines. It is also known in the context of which HLA the viral antigens are presented [184]. The mode of antigen presentation in the context of the HLA may determine the nature and effectiveness of the immune response against HPV. In the case of HPV16, the E5 protein has been shown to reduce HLA class I expression specifically on the surface of infected cells, hindering their recognition by CD8+ T cells, which is one mechanism of immune evasion. E5 acts selectively by reducing the expression of HLA-A and B but not HLA-E. Antigen presentation in the context of HLA-E can lead to suppression of the immune response including NK cell function [185]. On the other hand, certain MHC alleles including HLA-DRB1*1501 and, HLA-DQB1*0602 have been shown to be associated with a higher susceptibility to persistent HPV infection and a higher incidence of cervical cancer. In addition, it appears that the presence or absence of an effective immune response may be influenced by the presence of different protein variants in the HPV type, either due to different genetic subtypes or due to differences in transcription and translation levels. T_reg_ lymphocytes may play an important role in suppressing the immune response against HPV. Persistent genital HPV infection, including HPV 16, as well as cervical cancer and enlarging and recurrent genital warts after treatment correlate with an increased number of T_reg_ lymphocytes, suggesting the involvement of these cells in the maintenance of HPV infection. It has been shown that the E7 protein of HPV16 can induce these cells. On the other hand, the presence of T_reg_ is a favorable prognostic factor in patients with HPV-associated oral cancer, which may mean that the role of this lymphocyte subpopulation is not unequivocal and may differ between HPV-associated diseases [186,187,188]. HPV can also affect the immune system by modifying the expression of certain host genes. The E7 protein of oncogenic HPVs increases the activity of the host DNA methyltransferase DNMT1. In this way, HPVs affect the host immune system by increasing DNA methylation, leading to decreased expression of certain genes, e.g., of the chemokine CXCL14. Another mechanism by which oncoprotein E7 acts on the expression of host genes important for the immune response is through histone modification. Thus, among other things, the production of TLR9 is inhibited. Furthermore, E6 and E7, by inhibiting K310 acetylation of p65, can inhibit signal transduction by the transcription factor NF-κB which can, among other things, inhibit the production of pro-inflammatory cytokines such as IL-1, IL2, IL-6, IL8 IL-12, and TNF-α [189].

Interestingly, despite the greater number of immune evasion mechanisms described, high-risk HPV viruses are more likely to be eliminated and less likely to cause lesions than low-risk viruses. A study of more than 600 female students found that 20% of those infected with HPV 16 or 18 developed CIN 2 and 6.7% developed CIN 3 within three years, while 64.2% of those infected with HPV6 or 11 developed genital warts [190].

The effectiveness of the immune response may also be influenced by the microenvironment and coexisting infections. Persistent HPV infection and intraepithelial neoplasia are more common in women with vaginal dysbiosis, *Gardnerella vaginalis* infection, and also with sexually transmitted diseases, especially *C. trachomatis* infection. Topical or oral probiotics are likely to play a protective role and even increase the likelihood of spontaneous resolution of CIN-type lesions [191,192,193].

## 8. HPV’s Association with Various Diseases

It has been confirmed that Beta, Gamma, Mu, and Nu HPV cause epidermal infections and Alpha HPV causes epidermal and mucosal epithelial infections. Beta and Gamma often cause asymptomatic infections, especially in immunocompetent individuals. Different Alpha types are associated with different diseases. The key questions in the pathogenesis of HPV infections are what determines the tropism of viruses to specific areas and, in particular, what determines that a particular virus is oncogenic. Undoubtedly, the genetic variability of the viral proteins has an influence, especially E4 in the case of site affinity and E5–E7 in the case of oncogenicity [159]. Moreover, it has been shown that different genetic variants of HPV 16 and 18 may differ in pathogenicity due to, among other things, differences in the action of the E6 protein [43]. However, the association of individual HPV types to specific locations or specific diseases is not an absolute rule as, for example, the HPV 2 that usually causes common warts as well as the carcinogenic HPV16 can be found in genital warts. Many different types of HPV have also been found in focal epidermal hyperplasia lesions. HPV 16 and 18, but not HPV 6 and 11, were found more frequently in cancers and colorectal polyps than in the healthy intestinal epithelium, which may suggest a possible involvement of oncogenic HPVs in the pathogenesis of these diseases [194].

The association of different types of HPV with selected skin or mucosal lesions is shown in Table 2.

Although Beta and Gamma HPVs usually cause asymptomatic infections and are sometimes considered part of the healthy skin microbiome, in the case of certain genetic conditions (such as mutations in the *EVER* genes) and in immunosuppressed individuals, they can cause skin lesions and, in the case of beta HPVs, also contribute to the development of non-melanoma skin cancers (NMSCs). There is an increasingly well-founded hypothesis that the above-mentioned groups, but also to some extent in healthy individuals, Beta HPV infection is a factor in the formation of NMSCs under the effect of sun exposure. Exposure to ultraviolet radiation causes immunosuppression, which favors HPV persistence. Interestingly, Beta HPV DNA was found more frequently in the UV-exposed epidermis. On the other hand, HPV infection has a negative effect on the repair mechanisms and regulation of cell proliferation in the UV-exposed epidermis. HPV Beta DNA is detected in 30–50% of NMSCs in immunocompetent individuals and up to 90% of NMSCs in immunosuppressed individuals. HPV 5 and 8 are detected in nearly 90% of skin cancers in EV patients [10,183,199,200]. It is additionally worth noting that, according to the hit-and-run theory, the absence of the virus in tumor lesions does not completely rule out its involvement in pathogenesis. On the other hand, it is possible to speak of a so-called innocent bystander, a situation in which the pathogen is more frequently detected in tumor lesions than in healthy tissue, but is not the cause of the lesions [179].

## 9. Prophylactic Vaccines

All available HPV prophylactic vaccines contain recombinant L1 proteins of HPV, spontaneously forming virus-like particles (VLPs). The first vaccine, Gardasil, containing the L1 proteins of HPV 6, 11, 16, and 18 was approved by the FDA in 2006. Another vaccine is Cervarix, containing HPV 16 and 18. Cervarix has been shown to induce higher antibody titers than Gardasil, which does not seem to affect the effectiveness of vaccines against the viruses whose antigens they contain. However, this may affect the likelihood of cross-protection against certain types of oncogenes not covered by vaccines. As shown, cross-protection against HPV 31 was 77.1%, and 42.6% and against HPV 45 was 79% and 7.8% for bivalent and quadrivalent vaccines, respectively [201]. In 2014, Gardasil-9 against HPV 6, 11, 16,18, 31, 33, 45, 52, and 58 was introduced. Until 2015, the majority of HPV vaccines administered in the United States were 4vHPV vaccines. As of 2016, the 4vHPV vaccine is no longer available in the US and has been completely replaced by the 9vHPV vaccine [202].

According to the clinical studies, the efficacy of Gardasil, Gardasil-9, and Cervarix in preventing cervical, vulvar, and penile intraepithelial neoplasia (CIN, VIN, and PIN) caused by the viruses covered by the vaccine is more than 90% (approaches 100% in some studies and groups). In the case of Gardasil/Gardasil-9, the effectivity is 89–98% for genital warts. A study in sexually active adolescent girls and women, from different ethnic groups, aged 14–24 in the United States showed that the prevalence of HPV types covered by the 4-valent vaccine decreased by more than 80% between 2015 and 2018, compared to the years before the introduction of the vaccine 2003–2006. This indicates that the vaccine is highly effective in preventing HPV infections and, presumably, also cancers caused by these viruses. There was also a decrease in the incidence of new cases of CIN 2 and cervical cancer, which was greatest (by 29% in 2011–2015 compared to 2003–2006) in the youngest age group (15–24 years). A study in Australia, where approximately 73% of adolescent girls had been vaccinated by 2010, showed a decrease in condyloma incidence in the population aged 15 to 30. The largest decrease was in women under 21 years of age (from 11.5% in 2007 to 0.9% in 2011) [203,204].

The development of cheaper methods of producing vaccines has allowed their increased use. In recent years, three more vaccines have been registered, two in China (HPV 16 and 18) and one in India (HPV 6, 11, 16, and 18). Phase III clinical trials have been completed in China for a further two vaccines (HPV 6, 11, 16, and 18, and HPV 6, 11, 16, 18, 31, 33, 45, 52, and 58). The non-inferior HPV type-specific immune response to that of Gardasil and acceptable safety profiles of the above vaccines have been shown. All of the above vaccines have a much greater capacity to induce an immune response, including the production of blocking antibodies and cellular immunity, than natural infections [205,206]. The current prophylactic vaccines registered in various countries are shown in Table 3.

Another direction of investigation involves the development of prophylactic vaccines based on the use of the L2 protein. L2 could provide broad-spectrum protection against different HPV types due to its high preservative content. However, L2 has lower immunogenicity than L1 and has no ability to form virus-like particles on its own. VLPs L1 HPV16, VLP bacteriophage MS2 VLP-16L2 (aa17-31), and adeno-associated virus AAVLP-HPV, among others, are used as L2 carriers to improve its properties. Another possibility is to conjugate L2 with immunostimulating agents, e.g., *Pyrococcus furiosus* thioredoxin. There are several L2-based vaccines at the in vitro and animal trial stage that are likely to enter the clinical trial phase in the near future. At the clinical trial stage are some peptide vaccines containing L2 or prophylactic-therapeutic vaccines containing L2 conjugated to E6-E7 oncogenic HPVs [207,208].

## 10. Therapeutic Vaccines

The question arises whether currently approved prophylactic vaccines can be used as therapeutic vaccines. A study performed in a group of women after cervical conization for HSIL-type lesions showed that 4.3% of those who received bivalent or quadrivalent vaccine and 9.8% of unvaccinated women had recurrent lesions. The difference was statistically significant. A further study showed 58.7% efficacy of quadrivalent HPV vaccine in women with CIN 1-3. In another study, women with CIN 1-3 were vaccinated with a bivalent vaccine showed a vaccine efficacy 60 days after treatment of 88.2% for CIN 2-3 and 42.6% for CIN 1. There are also studies that have found a reduced risk of recurrence of anal intraepithelial neoplasia (AIN) in MSM and recurrence of vulvar lesions. None of the studies reported serious adverse reactions to the vaccines. All these results encourage the use of the vaccines in patients after the removal of lesions caused by HPV. On the other hand, a meta-analysis showed only a probable reduction in recurrence after surgical removal for CIN, especially CIN 2-3 caused by HPV 16 and 18, and insufficient evidence of such a relationship for other HPV-caused lesions. This may be due to too small study groups in the latter case and requires further research. Even more so, there is no evidence of HPV elimination or cessation of HPV-induced lesions by prophylactic vaccines. The mechanism of action of vaccines in women with previously removed HPV lesions is not fully understood, and in particular it is uncertain whether it consists only in protection against reinfection or also in the elimination of existing infection. There is no doubt that, as in the case of uninfected people, anti-L1 antibodies are produced to block the entry of HPV into cells. However, this highly effective mechanism does not work on viruses already inside cells. Vaccine-induced cell-mediated immunity may also play a role [209,210,211,212].

Therapeutic vaccines most often contain E6 and/or E7, and possibly E5 or E2 in the form of proteins or DNA or RNA encoding them. The problem is the low immunogenicity of these proteins and their ability to inhibit the host immune response. HPV genes/proteins contained in therapeutic vaccines are usually modified to increase immunogenicity and to eliminate oncogenic and immunosuppressive properties. The therapeutic vaccines can be divided into: (1) those containing a small DNA molecule (plasmid), (2) vector vaccines using other microorganisms (viruses and bacteria) to deliver HPV DNA, (3) RNA vaccines, (4) peptide vaccines, and (5) vaccines using autologous cells of the host immune system that present HPV antigens. A number of therapeutic HPV vaccines are currently in clinical trials. The trials mostly involve patients with HPV-dependent premalignant lesions and cancers, but there are also attempts to use some of them for persistent, recurrent benign lesions [213,214].

Table 4 shows selected therapeutic vaccines undergoing clinical trials used as a monotherapy or in combination with other vaccines in HPV-infected patients with precancerous or benign lesions.

These are mainly DNA vaccines in the form of plasmids or in viral and bacterial vectors, although there was also a first-phase study on peptides. Methods of vaccine administration and number of doses varied, which could also affect efficacy. Phase III clinical trials for the use of vaccine VGX-3100 containing two plasmids encoding optimized E6 and E7 of HPV16 and 18 in patients with CIN 1/3 showed resolution of cervical lesions in 23.7%. Another DNA vaccine is GX-188E containing plasmid encoding modified E6/E7 and Fms-like tyrosine kinase-3 ligand (Flt3L). II phase clinical trials. pBI-11, pNGVL4a-CRT/E7(detox) [184,226]. MVA E2 is a vector vaccine containing E2 bovine papillomavirus (BHV) in vaccinia virus. Due to the high similarity of this protein between different papillomaviruses, cross-reactivity is expected. Phase III studies have shown resolution of cervical intraepithelial neoplasia-type lesions in 89% of vaccinated women and elimination of the virus in 81%, a very promising result [184].

There are also numerous studies on patients with HPV-induced cancers. Here, therapeutic vaccines (including many of those listed in Table 4 are usually used in combination with chemotherapy or other cancer drugs such as immune checkpoint inhibitors (ICIs). Research concerns cervical cancer as well as HPV-caused head and neck cancers of the vulva, penis and anus. The studies which will not be discussed in detail here. Of the vaccines not mentioned earlier, studies include the use of MEDI0457/INO-3112, a vaccine containing 3 plasmids encoding the genes for E6 and E7 of HPV 16 and 18 and as an adjuvant the gene for IL-12 for the treatment of head and neck cancers and recurrent/metastatic cervical, anal, and penile cancers [227,228]. Another studies involved the ADXS11-001 LM-LLO vaccine containing attenuated *Listeria monocytogenes*-designed to secrete an antigen-adjuvant fusion protein consisting of a truncated fragment of *Listeria monocytogenes* listeriolysin O—LLO fused to HPV16 E7. An I/II phase clinical trial in patients with HPV-associated cancers showed encouraging results. A Phase III trial in patients with cervical cancer is currently underway. The likely mechanism of action is that the bacterium infects antigen-presenting cells, produces the HPV-LLO protein inside them, which is then presented to cytotoxic T cells [229].

Vaccines based on autologous immune cells modified to present HPV antigens seem to be a very interesting solution. However, their wider introduction is currently practically impossible due to the need to produce the vaccine individually for each patient, which significantly increases the price and makes it difficult to control vaccine quality. In 2008, the safety and immunogenicity of a vaccine containing autologous dendritic cells presenting E7 HPV 16 and 18 was demonstrated in phase IB/IIA [207].

At the moment, RNA vaccines for the treatment of HPV infection are still in the preclinical testing stage. Recently, the results of preclinical study of 3 capable or incapable self-replicating therapeutic vaccines containing lipid nanoparticle (LNP)-encapsulated mRNA for a chimeric protein resulting from the fusion of E7 HPV and HSV1 glycoprotein D were published. Demonstrated immunogenicity and efficacy on an animal model [230].

## 11. Drugs Currently Used in the Treatment of HPV Infection

Numerous surgical methods are used to treat HPV infections, including liquid nitrogen cryotherapy, electrocoagulation, laser therapy, curettage, surgical excision, trichloroacetic acid and photodynamic therapy-which which are beyond the scope of this article. The non-surgical treatment of HPV-induced lesions include topical application of substances with cytotoxic effects or stimulating the host immune system to eliminate the virus.

Preparations containing salicylic acid or a combination of salicylic acid and 5-fluorouracil are also often used in the treatment of cutaneous warts. Topical or intralesional 5-fluorouracil alone can be used in various HPV caused lesions. The latter drug inhibits thymidylate synthase which leads to inhibition of DNA replication. An additional mechanism is incorporation into DNA and RNA, which leads to inhibition of their synthesis. All these mechanisms ultimately lead to the death of rapidly dividing cells, e.g., tumor cells or HPV-infected cells. Topical 5 fluorouracil is available as a 5% carcinomas cream or in combination with 10% salicylic acid as a 0.5% solution [231,232].

Topical cantharidin and also the combination of 30% salicylic acid, 1% cantharidin and 5% podofixiloxin are sometimes used in the treatment of dermal warts, especially refractory foot warts. Another therapeutic option is intralesional injection of bleomycin, which has antimitotic and cytotoxic effects on rapidly dividing cells mainly through induction of DNA strand breaks [231,232,233].

Podophyllotoxin is an antimitotic compound isolated from the roots and rhizomes of plants belonging to the *Podophyllum* species, such as *Podophyllum peltatum* and *Sinopodophyllum hexandrum Royle*. Podophyllotoxin binds to the tubulin subunit of spindle microtubules, inhibiting their polymerization and producing an antimitotic effect that causes cell cycle arrest at metaphase. Podophyllotoxin may cause local tissue necrosis, which can halt HPV activity. Podophyllotoxin is used topically to treat genital warts in the form of a 0.5% solution (FDA approved in 1990) or gel, or a 0.15% cream [231].

Imiquimod is a nucleoside analogue and acts as an agonist for TLR 7, an intracellular PRR whose natural ligand is single-stranded RNA. Through TLR7, imiqumod acting on various intracellular signal transduction pathways leads to increased production of numerous cytokines, including IL 1, 2, 6, 8, 12, 18 and, most importantly, with regard to antiviral activity, also IFN α,β and γ. In addition, imiquimod interferes in a TLR-independent manner with adenosine receptor signaling pathways and reduces adenylyl cyclase activity, leading to negative feedback inhibition and maintenance of the inflammatory response. Finally, high concentrations of imiquimod have the ability to induce apoptosis through the activation of caspase. Imiqiumod in the form of a 5% cream was registered by the FDA in 1997 and is recommended for the treatment of anogenital warts and solar keratosis, Bowen’s disease and superficial basal cell carcinomas. It is also used to treat lesions other than genital warts caused by HPV, including vulvar, penile, anal and cervical intraepithelial neoplasia, as well as skin warts [234,235,236].

Sinecatechins are an aqueous extract of green tea (Camellia sinensis) leaves containing a mixture of several different substances, mainly polyphenols of which epigallocatechin gallate (EGCG) is the most abundant (>65%). Others include epicatechin (>10%), epigallocatechin (<10%), epicatechin gallate (<10%) and the remaining catechins and other compounds present in smaller amounts (<10%). Studies on the mechanism of action of sinecatechins have focused on epigallocatechin gallate. This action is multidirectional and includes inhibition of E6 and E 7 HPV expression, activation of caspase, inhibition of telomerase activity altered Bcl-2 expression and in consequence inhibition of proliferation (cell cycle arrest in G0/G1 phase) and induction of apoptosis. It also inhibits angiogenesis by suppressing STAT3 activation. EGCG also inhibits DNA methyltransferase leading to increased expression of certain genes including interferon-stimulated genes which may be associated with antiviral and antitumor effects, as well as immunomodulatory effects, among others. Antioxidant activity may also be important, both directly by reducing reactive oxygen and nitrogen species and indirectly by inhibiting inflammatory reactions (e.g., NF-kB pathway, AP-1, as well as cyclogenases and lipogenases). ECGC also stimulates the production of antioxidant enzymes such as superoxide dismutase, catalase and glutathione. An ointment containing 15% standardized green tea extract (Polyphenon^®^ E) was approved by the FDA in 2006 for the treatment of anogenital warts. Cases of successful use in other HPV-caused lesions like facial warts and foot warts have been described [232,237].

Cidofovir (CDV) is an analogue of cytosine monophosphate, which is converted to diphosphate under the influence of a cellular kinase and then incorporated into viral DNA, inhibiting its synthesis. In the case of viruses possessing their own DNA polymerase, the mechanism of action is a competitive inhibition of this enzyme. However, in the case of HPV using cellular DNA polymerase, one can speak more of causing the death of infected cells as a result of apoptosis cells than of a direct antiviral effect. The drug in topical (usually at a concentration of 1–3%) or intralesional form is used to treat a variety of lesions caused by HPV, but its availability is limited [236,238].

An interesting option for the treatment of HPV-caused lesions is intralesional non-specific immunotherapy using candida antigens, Measles, Mumps, and Rubella (MMR) vaccine, Bacillus Calmette-Guérin (BCG) or tuberculin purified protein derivative (PPD). The likely mechanism of action is to stimulate PRRs especially TLRs and consequently stimulate the production of various cytokines especially interferons. Intralesional immunotherapy is mainly used in refractory common warts, but there are also descriptions of use in genital warts. For treatment of common warts, the percentage of people whose lesions resolved completely in the various groups studied was 26–92% for MMR, 39–88% for candida antigens, 23.3–94.4% PPD and 33.3–70% for BCG [195,239].

Another method recently reported to be effective in refractory HPV-positive lesions is intralesional injection of vitamin D. In this case, the likely mechanism of action involves the regulation of keratinocyte proliferation and, at the same time, the inhibition of the production of certain cytokines such as interleukin-6 (IL-6), IL-8, and tumor necrosis factor (TNF)-α and the stimulation of the production of IFN gamma. A study of patients with common warts showed higher efficacy of intralesional vitamin D than intralesional BCG [240].

## 12. New Drug Therapy

Most research on new HPV drugs is still at the in silico and in vitro stage.

E1 is one of the few HPV proteins showing enzymatic activity, making it a potential target for new drugs. It is also characterized by low variability between HPV types. In recent years, in silico studies have been carried out to find potential E1 inhibitors among the already known substances included in the drug database. The best results were obtained with Cinalukast, Lobeglitazone, and Efatutazone. The authors suggest that the chemical structures of these substances could provide a basis for the design of E1 inhibitors [241]. In earlier studies, small-molecule chemical indandione has been shown to be able to inhibit the interaction between E1 and E2 HPV, raising hopes for its future use as an anti-HPV drug [242]. Several substances have also been identified that inhibit HPV 16 and 18 replication in vitro by inhibiting the cellular enzymes poly [ADP-ribose] polymerase 1-PARP1 and tyrosyl-DNA phosphodiesterase 1-Tdp1 [243]. Another potential group of drugs that inhibits in vitro the initiation of HPV replication by acting on E2 or E1–E2 are pyrrole-imidazole polyamides [244].

There are also in silico in vitro studies on the inhibition of ion channel formation by E5 proteins by alkylated amino sugars. Moreover, there is an inhibition of some of the metabolic pathways stimulated by E5 and, as a consequence, a decrease in cell proliferation and stimulation of cell differentiation [245].

Numerous studies address the inhibition of E6 and E7 both by blocking the production of these proteins and inhibiting their action. Some focus on inhibiting the expression of these proteins by small interfering RNA (siRNA), or antisense deoxynucleotides. Other studies are investigating the inhibition of these proteins and their actions at the post-transcriptional stage. This can be achieved by various means, including the use of ribozymes, intrabodies targeting E6 proteins, flavonoid-derived compounds, peptides, small-molecule substances, and others [244]. One study using Structure-Based Virtual Screening (SBVS) identified a potential small-molecule drug that inhibits the binding of the of HPV18 E7 and the cellular PTPN14 (Protein Tyrosine Phosphatase Non-Receptor Type 14). This results in inhibition of PTPN14 degradation and increased levels of PTPN14, resulting in the inhibition of Yes-associated protein (YAP) and the inhibition of cell proliferation. The substance may be used to treat cancers caused by HPV [246].

Interesting results were also obtained with 3-hydroxyphthalic anhydride (3HP)-modified bovine β-lactoglobulin (3HP-β-LG). This protein was shown to bind to the L1 positively charged fragment of HPV and block the entry of viral particles into cells with high efficiency. Potentially in the future, the substance could be used for the topical treatment of HPV infections [247]. Other in vitro studies of L1 HPV 16 virus-like particles have shown that non-functionalized gold nanoparticles (nfGNPs) strongly inhibit viral entry into cells [248].

Other promising in vitro studies have looked at the action of a 29-amino-acid peptide designated P16/16, containing the CPP (cell-penetrating peptide) and the RBS (retromer-binding site) from HPV16 L2. The first sequence allows penetration into cells, and the second binds to a cytoplasmic retromer protein that allows HPV transport through the Golgi apparatus into the cell nucleus. This blocks of the association of L2 HPV with the retromer and inhibits the transport of the virus [249]. Other preclinical studies in vitro and in a mouse model have shown that protamine sulfate probably combines with heparan sulphate to block virus attachment and entry into the cell interior. Clinical trials are likely to be initiated [250].

In addition to the previously mentioned epigallocatechin derived from Camelia sinensis, numerous other substances of plant origin have been shown to inhibit HPV infection in vitro or in vivo including propolis, reservatrol, curcumin, silymarin, neem, and berberine. This could lead to the development of new drugs [251,252].

## 13. Conclusions

In order to make HPV systematics clearer, it would be necessary to merge and unify the information contained in the various databases;Current treatment is mainly based on:-Surgical procedures;-Topical or intralesional application of substances with antiproliferative and cytotoxic effects on infected cells (e.g., podophyllotoxin, bleomycin, 5-fluorouracil, cidofovir) or non-specific stimulation of the immune system to destroy HPV (e.g., imiquimod, intralesional immunotherapy). Some of the drugs used such as sinecatechins and vitamin D have both immunostimulating and antiproliferative effects;A number of therapeutic vaccines (specific immunotherapy) are undergoing clinical trials, some of which, such as MVA E2 and VGX-3100, have completed phase III clinical trials and are expected to be available soon;It is encouraging that prevention of infections with the most common mucosal HPV types is available. The efficacy of vaccines based on virus-like particles from the L1 protein is fully proven. Also noteworthy is the effect of prophylactic vaccines on reducing recurrence in women treated for CIN. These vaccines are increasingly used worldwide. However, in order to reduce the number of infections and their impact, it would be advisable to further increase the availability of vaccines and to disseminate more effectively information regarding their efficacy and safety;Despite a great deal of research, there are still no drugs available that specifically inhibit HPV replication. However, an increasing understanding of the HPV replication cycle and the structure and function of individual viral proteins should, in the future, allow the development of effective, specific pharmacotherapy. Research on such drugs is mostly still in the preclinical phase;The treatment of precancerous lesions, cancers and persistent/recurrent benign lesions caused by HPV is still an incompletely solved problem. The effectiveness of treatment may be enhanced by the inclusion of methods which are based on the stimulation of the immune system in the fight against infection.

## Figures and Tables

**Figure 1 ijms-25-07616-f001:**
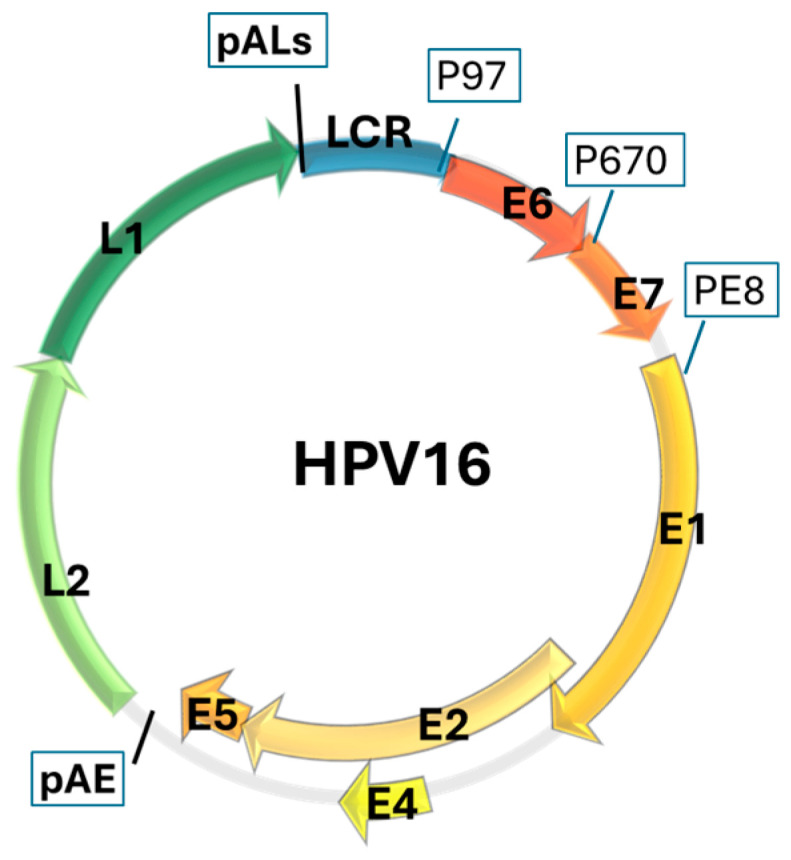
HPV genome E (early)—orange, L (late)—green gens, LCR (long control region)—blue, P97, P670, PE8—promoters, pAE—early polyadenylation sites, pALs—late polyadenylation sites [158].

**Figure 2 ijms-25-07616-f002:**
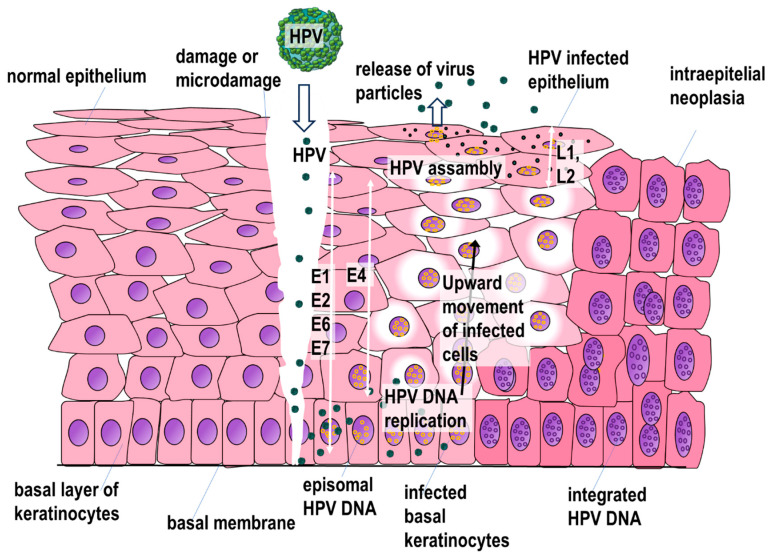
The life cycle of HPV. White arrows—expression of early E1, 2, 4, 6, 7, and late L1, 2 HPV genes.

**Table 1 ijms-25-07616-t001:** Classification of HPVs. High-risk mucosal HPVs are circled in yellow. HPVs classified as oncogenic and probably oncogenic are not underlined; those which are possibly oncogenic are underlined in green. Low-risk mucosal HPV types are circled in green. In gray with green underlining are HPV types classified as possibly oncogenic in patients with EV [16]. ICTV—International Committee on Taxonomy of Viruses, TB—Taxonomy Browser, IRHC—International Human Papillomavirus Reference Center, PaVe—The Papilloma Virus Episteme.

Viruses (Superkingdom); Monodnaviria (Clade); Shotokuvirae (Kingdom); Cossaviricota (Phylum); Papovaviricetes (Class); Zurhausenvirales (Order); Papillomaviridae (Family); Firstpapillomavirinae (Subfamily)	
**genus**	** Alphapapillomavirus **
species	**Alphapapillomavirus 1** [17,18,19]
type	HPV32, 42
species	**Alphapapillomavirus 2** [18,20,21,22,23,24]
type	HPV3, 10, 28, 29, 77, 78, 94, 117, 125, 160; the TB [13] database also includes HPVXS2
species	**Alphapapillomavirus 3** [17,25,26,27,28,29,30,31,32,33]
type	HPV61 sublineage A1, A2, lineage B, C, HPV62, 72, 81, 83, 84, 86, 87, 89, 102, 114
species	**Alphapapillomavirus 4** [18,34,35,36]
type	HPV2, lineage 2a and 2c; HPV27, lineage b; HPV57, lineage b and c
species	**Alphapapillomavirus 5** [18,26,37,38,39]
type	HPV26 lineage A, HPV51 sublineage A1 to A4, B1, B2, HPV69 sublineage A1 to A4, HPV82 sublineage A1 to A3, B1, B2, C1 to C5
species	**Alphapapillomavirus 6** [18,26,40,41]
type	HPV30: sublineage A1 to A3 and lineage B, HPV53: lineage A, B, C, and sublineage D1 to D4
	HPV56: sublineage A1, A2, and lineage B; HPV66: lineage A and sublineage B1, B2
species	**Alphapapillomavirus 7** [18,26,39,42,43,44,45,46,47,48,49,50,51,52]
type	HPV18 sublineage A1 and A2 (Asian-Amerindian), A3 to A5 (European), B1 to B3 (African), and lineage C (African)
	HPV39 sublineage A1, A2, and lineage B; HPV45 sublineage A1 to A3, B1, B2
	HPV59 sublineage A1 to A3, B, and lineage B
	HPV68 lineage a, sublineage A1, A2, lineage B, and b, sublineage C1, C2, sublineage D1, D2, lineage E, and sublineage F1, F2
	HPV70 lineage A and B; HPV85 lineage A; HPV97 lineage A
species	**Alphapapillomavirus 8** [17,18,29]
type	HPV7; HPV40; HPV43; HPV91
species	**Alphapapillomavirus 9** [18,26,43,53,54,55,56,57,58,59,60,61,62,63,64,65,66,67,68,69]
type	HPV16 sublineage A1 to A3 (European), A4 (Asian), B1 (African-1, Afr1a), B2 African-1, Afr1b), B3 and B4, C1 (African-2, Afr2a) AF472509, C2, C3, C4, D1 (North American, NA1), D2 (Asian-American, AA2), D3 (Asian-American AA1), and D4
	HPV31: sublineage A1, A2, B1, B2, C1 to C4; HPV33: sublineage A1 to A3, B1, C1
	HPV35: sublineage A1, A2; HPV52: A1, A2, B1, B2, B3, C1, C2, D1, E1
	HPV58: sublineage A1 to A3, B1, B2, C1, D1, D2; HPV67: sublineage A1, A2, B1
species	**Alphapapillomavirus 10** [17,70,71,72,73,74,75,76,77,78,79,80,81,82,83,84,85]
type	HPV6: lineage A, 6a (classified as sublineage B3), 6b (classified as lineage A, 6c, 6e, 6vc (classified as sublineage B1), and sublineage B1, B2, B3, B4, B5
	HPV11: sublineage A1, A2, A3 (isolate C185), A4 (isolate LT4), and lineage B; HPV13; HPV44; HPV74
	The TB database [13] also includes HPV55
species	**Alphapapillomavirus 11** [18,26,86]
type	HPV34 sublineage A1, A2, lineage B, and sublineage C1, C2
	HPV73 sublineage A1, A2, and lineage B
	In addition, the TB database [13] includes HPV177
species	**Alphapapillomavirus 13** [26,39]
type	HPV54 lineage A, B, C
species	**Alphapapillomavirus 14** [29,30,39]
type	HPV71; HPV90; HPV106
**genus**	** Betapapillomavirus **
species	**Betapapillomavirus 1** [18,87,88,89,90,91,92,93,94,95,96,97]
type	HPV5: lineage b; HPV8, 12; HPV14: lineage D; HPV19, 20, 21, 24, 25, 36, 47, 93, 98, 99, 105, 118, 124, 143, 152
	In addition, the [13] database includes HPVRTRX7, HPVV001/Slovenia/2010
	In addition, the [14] database includes HPV195, 196, 206, which in ICTV [12] appear as unclassified Betapapillomavirus HPVmRTRX7
species	**Betapapillomavirus 2** [18,20,90,98,99,100,101,102,103,104,105,106,107,108,109]
type	HPV9, 15, 17, 22, 23, 37; HPV38: lineage b; HPV80, 100, 104, 107, 110, 111, 113, 120, 122, 145, 151, 174
	In addition, the [14] database includes HPV182, 198, 209, 227
	In addition, the [13] database includes HPVFA75/KI88-03
species	**Betapapillomavirus 3** [18,110]
type	HPV49, 75, 76, 115
species	**Betapapillomavirus 4** [111]
type	HPV92
	**Betapapillomavirus 5** [93,102,112]
	HPV96, 150
	In addition, the [13,14] databases include HPV185
**genus**	** Gammapapillomavirus **
species	**Gammapapillomavirus 1** [113,114,115]
type	HPV4, 65, 95,173, 205
species	**Gammapapillomavirus 2** [116,117]
type	HPV48 {NC_001690}; HPV200
species	**Gammapapillomavirus 3** [118]
type	HPV50
	In addition, the [13,14] databases include HPV188
species	** Gammapapillomavirus ** **4**
type	HPV60 [119]
species	** Gammapapillomavirus ** **5**
type	HPV88 [120]
species	**Gammapapillomavirus 6** [121,122,123,124]
type	HPV101, 103, 108
	In addition, the [14] database includes HPV214, 226
species	**Gammapapillomavirus 7** [33,95,96,125,126,127,128]
type	HPV109, 123, 134,138,139,149, 155, 170
	In addition, the [13,14] databases include HPV186, 189, 193
	Furthermore, the [14] database includes HPV203, 225, 229
species	**Gammapapillomavirus 8** [33,95,123,127,129]
type	HPV112, 119, 147,164, 168
	Furthermore, the [13,14] databases include HPV176
	In addition, the [14] database includes HPV211, 224, which in ICTV [12] appear as unclassified Betapapillomavirus HPVmICB1.
species	**Gammapapillomavirus 9** [123,126,130]
type	HPV116, 129
	In addition, the [14] database includes HPV 215, 216
species	**Gammapapillomavirus 10** [95,126,131,132]
type	HPV121,130, 133, 142, 180
	In addition, the [13,14] databases include HPV191
	Furthermore, the [14] database includes HPV221, 231
species	**Gammapapillomavirus 11** [96,114,117,127,133,134]
type	HPV126, 136, 140, 141, 154, 169, 171, 202
	In addition, the [13,14] databases include HPV181
	Furthermore, the [13] database includes HPV230
species	**Gammapapillomavirus 12** [115,126,127,135,136,137]
type	HPV127, 132, 148, 157, 158, 165, 199
	In addition, the [13,14] databases include HPV210
species	**Gammapapillomavirus 13** [123,126,132,138]
type	HPV128, 153
	In addition, the [14] database includes HPV213, 219
species	**Gammapapillomavirus 14** [126]
type	HPV131
species	**Gammapapillomavirus 15** [96,114,139,140]
type	HPV135, 146, 179
	In addition, the [13,14] databases include HPV192 a
	In addition, the [14] database includes HPV230
species	**Gammapapillomavirus 16** [96]
type	HPV137
species	**Gammapapillomavirus 17** [96,123,132]
type	HPV144
	In addition, the [14] database includes HPV212, 220
species	**Gammapapillomavirus 18** [141]
type	HPV156
species	**Gammapapillomavirus 19** [127,132]
type	HPV161, 162, 166
	In addition, the [14] database includes HPV222
species	**Gammapapillomavirus 20** [127]
type	HPV163
	In addition, the [13,14] databases include HPV 183 a
	Furthermore, the [14] database includes HPV194
species	**Gammapapillomavirus 21** [129]
type	HPV167
species	**Gammapapillomavirus 22** [114]
type	HPV172
	In addition, the [14] database includes HPV223
species	**Gammapapillomavirus 23** [131]
type	HPV175
species	**Gammapapillomavirus 24** [142,143]
type	HPV178, 197; in addition, the [14] database includes HPV190, 208
species	**Gammapapillomavirus 25** [139]
type	HPV184
species	** Gammapapillomavirus ** **26**
type	HPV187
species	**Gammapapillomavirus 27** [117]
type	HPV201; in addition, the [14] database includes HPV228
**genus**	** Mupapillomavirus **
species	**Mupapillomavirus 1** [144]
type	HPV1; in the [13] database, it appears under the name HPV1a
species	**Mupapillomavirus 2** [113]
type	HPV63
species	**Mupapillomavirus 3** [145,146]
type	HPV204
**genus**	** Nupapillomavirus **
species	**Nupapillomavirus 1** [147]
type	HPV41

**Table 2 ijms-25-07616-t002:** The association of different types of HPV with selected skin or mucosal lesions.

Type of Skin/Mucosal Lesions	HPV Types	References
plantar warts	usually HPV2, 27, 57, 63, furthermore HPV 1, 4, 10, 41, 65, 88, 95, 60, 65, 66	[195,196]
common warts	HPV 27, 57,2,1,4	[159,196]
flat warts	HPV3, 10, 26, 27, 28, 29, 77, 78, 94, 114, 41	[196]
genital warts (*Condyloma acuminatum*)	usually (90%) HPV 6 and 11, less commonly HPV 2, 16, 18, 30–33, 35, 39, 41–45, 51–56, and 59	[159,197]
cervical intraepithelial neoplasia, and cancer	HPV 16 (mostly), HPV18, 31, 33, 35, 39, 45, 51, 52, 56, 58, 59 (carcinogenic), 68 (probably carcinogenic), 26, 53, 66, 67, 69, 70, 73, and 82 (possibly carcinogenic)	[16]
focal epithelial hyperplasia	mainly HPV13 and 32, but infection or co-infection with other HPVs including HPV6, 11, 16, 18, 31, 39, 40, 51, 52, 55, 58, 66, 68, 69, 71, 74, and 90 has also been reported	[198]
warts and probably NMSC in EV and immunocompromised patients; in the general population usually asymptomatic but also may be associated with NMSC	HPV5, 8 (possibly carcinogenic) HPV9, 12, 14, 15, 17, 19–25, 36–38, 47, 49, 75, 76, 80, 92, 93, 96, 98–100, 104, 105, 107, 110, 111, 113, 115, 118, 120, 122, 124, 143, 145	[199]

**Table 3 ijms-25-07616-t003:** Currently available prophylactic HPV vaccines.

Vaccine (Manufacturer)	HPV Types Included	Adjuvant	Method of Producing HPV Proteins	Registration Year	Vaccine Schedule
Gardasil^®^ (Merck & Co., Rahway, NJ, USA)	6 (20 µg), 11 (40 µg), 16 (40 µg), 18 (20 µg) quadrivalent	Amorphous aluminium hydroxyphosphate sulphate (225 µg Al)	*Saccharomyces cerevisiae* (yeast) expressing L1	2006	I.M. 9–14 years: 0, 6 months from the age of 15 0, 2, 6 months
Cervarix^®^ (GlaxoSmithKline, Tsim Sha Tsui, Hong Kong)	16 (20 µg), 18 (20 µg), bivalent	AS04: 3-*O*-deacylo-4′-monofosforylolipid A (MPL) 50 μg, adsorbed on aluminium hydroxide (0.5 mg Al^3+^)	Utilizing a baculovirus expression system, with the use of insect Hi-5 Rix4446 cells derived from *Trichoplusia ni.*	2007	I.M. 9–14 years: 0, 6 months from the age of 15 0, 1, 6 months
Gardasil9^®^ (Merck & Co.)	6 (30 µg), 11 (40 µg), 16 (60 µg), 18 (40 µg), 31 (20 µg), 33 (20 µg), 45 (20 µg), 52 (20 µg), 58 (20 µg) nonavalent	Amorphous aluminium hydroxy phosphate sulphate (0.5 mg Al)	*Saccharomyces cerevisiae* (yeast) expressing L1	2014	I.M. 9–14 years: 0, 6 months from the age of 15 0, 2, 6 months
Cecolin^®^ (Xiamen Innovax Biotechnology, Xiamen, China)	16 (40 µg), 18 (20 µg) bivalent	Aluminium hydroxide (208 μg Al)	*Escherichia coli* (bacteria) expressing L1	2020	I.M. 9–14 years: 0, 6 or 0, 1, 6 months from the age of 15 0, 1, 6 months
Walvax recombinant HPV vaccine (Hanghai Zerun Biotechnology, Shanghai, China; Subsidiary of Walvax Biotechnology, Shanghai, China)	16 (20 µg), 18 (40 µg) bivalent	Aluminium phosphate (225 µg Al)	*Pichia pastoris* expressing L1 (yeast)	2022	I.M. 9–14 years: 0, 6 or 0, 2, 6 months from the age of 15 0, 2, 6 months
Cervavac^®^ (Serum Institute of India, Pune, India)	6 (≥20 µg), 11 (≥40 µg), 16 (≥40 µg), 18 (≥20 µg) quadrivalent	Aluminium (Al^3+^) ≥ 1.25 mg	*Hansenula polymorpha* expressing L1 (yeast)	2022	I. M. 9–14 years: 0, 6 months from the age of 15 0, 2, 6 months

**Table 4 ijms-25-07616-t004:** Selected therapeutic vaccines undergoing clinical trials.

Vaccine Name (Manufacturer If Available)	Antigen/Other Vaccine Components	Type of Vaccine	Number of Administrations, Method of Delivery,	Research Stage	Results	References
TG4001 Tipapkinogen Sovacivec	DNA -modified HPV 16 E6 and E7 and human IL-2 in vaccinia virus Ankara (MVA)	DNA, virus-vectored	3 subcutaneous injections	phase II	CIN 2/3 lesions caused by HR HPV ceased in 24% of the 129 women who received the vaccine and in 10% of the 64 women receiving placebo. The result was statistically significant. The vaccine was well tolerated.	[215]
MVA E2	Cross-reactive E2 (bovine papilloma virus) in vaccinia virus Ankara (MVA)	DNA, virus-vectored	6 site-specific injections	phase III	In 1051 of 1176 women with CIN (89%) and in all 180 PIN men who received the vaccine, the lesions resolved completely. 81% of the women tested had eliminated oncogenic HPV.	[216]
				phase II	In 13 (43%) of the 29 patients studied, respiratory papillomatosis resolved completely; in the others, the lesions recurred between 8–12 months after vaccination, but resolved without further recurrence after re-administration.	[217]
VGX-3100 (Inovio Pharmaceuticals)	Modified E6 and E7 HPV-16 and -18 in two synthetic plasmids/pVAX	DNA plasmid	3 IM injections with electroporation	phase II NCT01304524	Regression of CIN 2/3 lesions confirmed by histopathological examination in 53 (49.5%) of 107 patients receiving the vaccine versus 11(30.6%) of 35 patients receiving placebo.	[218]
				phase III NCT03721978	In total, in the REVEAL 1 and 2 studies, cessation of CIN lesions and elimination of the virus was found in 68 of 272 (25.0%) women receiving the vaccine and in 13 of 132 (9.8%) receiving placebo.	[213]
GX-188E Tirvalimogene teraplasmid	Modified E6 and E7 HPV-16 and -18 in Plasmid/pGX27	DNA plasmid	3 IM injections with electroporation	Phase II	64 patients with CIN, histopathologically confirmed resolution of lesions in 52% (33/64) of patients after 20 weeks from the first administration of the vaccine and in 67% (35/52) after 36 weeks. HPV was eliminated in 73% and 77% of patients who had resolved lesions after 20 and 36 weeks, respectively.	[219]
pBI-11	pNGVL4a-Sig/E7(detox)/HSP70 plasmid encoding HPV16 L2E7E6 fusion protein	DNA plasmid	2 doses of pBI-11 + 1 dose of TA-CIN, I.M.	Phase II	HPV eradicated and lesions resolved after 6 months in 5 (45%) and after 12 months in 7 (64%) of 11 HPV16+ vaccinated women with ASC-US, ASC-H, or LSIL/CIN. Patients additionally vaccinated with the protein vaccine TA-CIN (recombinant HPV-16 L2 E6 E7).	[220]
GLBL101c	Heat-attenuated recombinant Lactobacillus casei expressing modified HPV16 E7	live-bacteria-vectored	4 rounds of oral vaccination, daily for 5 days at weeks 1, 2, 4, and 8	Phase II	No statistically significant differences in resolution of CIN 2/3 lesions between the 20 patients receiving the vaccine and the 20 patients receiving placebo. However, complete resolution of lesions in 2 patients in the vaccine group and none in the placebo group.	[221]
IGMKK16E7	Lacticaseibacillus paracasei expressing on cell surface, full-length HPV-16 E7	live-bacteria-vectored	4 rounds of oral vaccination daily at weeks 1, 2, 4, and 8.	Phase II	CIN 2/3 lesions resolved in 13 (31.7%) of 41 patients receiving the high dose of vaccine and in 5 (12.5%) of 40 patients receiving placebo. For patients infected only with HPV16, the incidence of resolution of lesions was 40.0% (12 of 30) in those receiving the vaccine and 11.5% (3 of 26) in those receiving placebo.	[222]
BLS-M07	Lactobacillus casei expressing on cell surface, full-length HPV-16 E7	live-bacteria-vectored	4 rounds of oral vaccination- daily for 5 days at weeks 1, 2, 4, and 8.	Phase I/IIa	Phase I—19 patients with HPV16 and CIN 3 infection—safe, immunogenic; phase IIa—lesions resolved in 6 of 8 patients studied.	[223]
HPV-16 E6 synthetic peptides	HPV-16 E6 synthetic peptides conjugated to Amplivant (optimized Toll-like receptor 2 ligand)	peptide	3 administrations, intradermally	Phase I	25 patients with HPV-16 positive (pre-)malignant lesions. Safe, immunogenicity increases with increasing dosage, but mild side effects (flu-like symptoms) are more frequent.	[224]
ISA 101	Overlapping long peptides 9 from the E6 HPV 16 protein and 4 from E7 in incoplete	peptide	3 or 4 administrations subcutaneously at 3-week intervals	Phase II	After 3 months response in 12 of 20 patients and complete regression of lesions in 5 of 20 patients. After 12 months, response in 15 of 19 patients with VIN 2/3 receiving the vaccine, complete regression of lesions in 9 of 15.	[225]

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
