# Peer review of "HPV Infections—Classification, Pathogenesis, and Potential New Therapies"

_ijms, 2024, doi:10.3390/ijms25147616_

Round 1

Reviewer 1 Report

Comments and Suggestions for Authors

The manuscript: "Human papillomavirus infections-from pathogenesis to potential new therapies" is a thorough review of the HPV pathogenesis, genotyping, prevention and therapy options. The introduction offers enough information for the reader and the chapters follow one another in a logical manner. Conclusion part focuses more on the therapy and prevention, while it should be a short overview of the article, starting from the viral genetics. Tables, especially 1 and 2 take substantial amount of text, while not contributing much to the better understanding of the manuscript, therefore I propose to the authors to place said tables as a supplement, and perhaps add just one, with a shorter overview of the HPV genotypes. When citing 3 or more consecutive references, it should be done in a like lines  form: [ 13-15], and it should be corrected in the text, such as line 81 for example. Other than that, I find the manuscript suitable for publishing.

Comments on the Quality of English Language

The English language requires moderate editing, perhaps the authors should have an English literature professor or a native speaker check the manuscript.

Author Response

Thank you for all your valuable comments

1.The conclusions have been expanded in accordance with the reviewer's recommendations

2.Table 1 has been shortened and the full version moved to the Supplementary

Table 2 has been moved to the Supplementary

3.Consecutive references have been corrected

4.Minor language errors and typos have been corrected in the text

Reviewer 2 Report

Comments and Suggestions for Authors

The review article written by Beata Mlynarczyk-Bonikowska and Lidia Rudnicka gives a detailed overview of the classification, pathogenesis, and potential therapies for human papillomavirus (HPV) infections. It emphasizes the role of viral proteins in disease development and explores future treatment prospects. The topic is very important as, so far, there is also no direct treatment for HPV infection. The manuscript is interesting and includes valuable information; however, I have some comments:

Major points:

  1. The manuscript's title is "Human papillomavirus infections—from pathogenesis to potential new therapies." However, a significant portion of this review discusses HPV classification, which should also be highlighted in the title.
  1. Tables 1 and 2 are too long and contain excessive information, which makes them difficult to follow.  
  1. In Chapter 5, "HPV replication cycle and pathogenesis of HPV-associated cancers," the authors describe the viral life cycle, then move to carcinogenesis and end it with a description of HPV capsid proteins and viral release, which do not happen during carcinogenesis. Can the author rearrange this chapter? It is also important to emphasize that E6 and E7 HPV proteins play different roles in the HPV life cycle and cancerogenesis process.
  1. Conclusions: The author could discuss more the practical challenges and limitations associated with current HPV therapeutic approaches, such as issues related to vaccine accessibility or the effectiveness of existing treatments.

Minor points:

  1. The description of Table 1 and Table 2 should be above the tables, not below them.
  2. Lines 141/143. Variable copies but less than 72 molecules (typically 20-40 molecules) of minor capsid protein L2 are incorporated within the viral particle (PMID: 18367526, 23689062).
  3. Line 142. The estimated molecular mass of HPV L2 is approximately 55KDa. However, due to some posttranslational modification, L2 typically exhibits an apparent molecular weight of 64-78kDa by SDS-PAGE analysis (PMID: 23689062).
  4. There are some typos and other errors. Additional proofreading is recommended.

Author Response

Thank you for all your valuable comments

The manuscript's title is "Human papillomavirus infections—from pathogenesis to potential new therapies." However, a significant portion of this review discusses HPV classification, which should also be highlighted in the title

1.The title has been changed in accordance with the reviewer's recommendations

   Tables 1 and 2 are too long and contain excessive information, which makes them difficult to follow. 

2.Table 2 has been moved to the Supplementary

Table 1 has been shortened and the full version moved to the Supplementary

In Chapter 5, "HPV replication cycle and pathogenesis of HPV-associated cancers," the authors describe the viral life cycle, then move to carcinogenesis and end it with a description of HPV capsid proteins and viral release, which do not happen during carcinogenesis. Can the author rearrange this chapter? It is also important to emphasize that E6 and E7 HPV proteins play different roles in the HPV life cycle and cancerogenesis process.

3.The text has been rearranged. The replication cycle and carcinogenesis are currently described in 2 chapters. This seems to be more clear.

Conclusions: The author could discuss more the practical challenges and limitations associated with current HPV therapeutic approaches, such as issues related to vaccine accessibility or the effectiveness of existing treatments.

4.The conclusions have been expanded in accordance with the reviewer's recommendations

    The description of Table 1 and Table 2 should be above the tables, not below them.

    Lines 141/143. Variable copies but less than 72 molecules (typically 20-40 molecules) of minor capsid protein L2 are incorporated within the viral particle (PMID: 18367526, 23689062).

    Line 142. The estimated molecular mass of HPV L2 is approximately 55KDa. However, due to some posttranslational modification, L2 typically exhibits an apparent molecular weight of 64-78kDa by SDS-PAGE analysis (PMID: 23689062).

5.Corrected according to the reviewer's recommendations, adding relevant literature items

    There are some typos and other errors. Additional proofreading is recommended

6.Typos and minor language errors have been corrected in the text